# A Semi-Lagrangian Method for Detecting and Tracking Deep Convective Clouds in Geostationary Satellite Observations

William K. Jones[1], Matthew W. Christensen[1], and Philip Stier[1]

[1]Atmospheric, Oceanic & Planetary Physics, Department of Physics, University of Oxford, Oxford, UK

**Correspondence:** William K. Jones (william.jones@physics.ox.ac.uk)

**Abstract.** Automated methods for the detection and tracking of deep convective storms in geostationary satellite imagery have a vital role in both the forecasting of severe storms and research into their behaviour. Studying the interactions and feedbacks between multiple deep convective clouds, however, poses a challenge for existing algorithms due to the necessary compromise between false detection and missed detection errors. We utilise an optical flow method to determine the motion of deep convective clouds in GOES-16 ABI imagery in order to construct a semi-Lagrangian framework for the motion of the cloud field, independently of the detection and tracking of cloud objects. The semi-Lagrangian framework allows for severe storms to be simultaneously detected and tracked in both spatial and temporal dimensions. For the purpose of this framework we have developed a novel Lagrangian convolution method and a number of novel implementations of morphological image operations that account for the motion of observed objects. These novel methods allow the accurate extension of computer vision techniques to the temporal domain for moving objects such as DCCs. By combining this framework with existing methods for detecting deep convective clouds (including detection of growing cores through cloud top cooling and detection of anvil using brightness temperature), we show that the novel framework enables reductions in errors due to both false and missed detections compared to any of the individual methods, reducing the need to compromise when compared with existing frameworks. The novel framework enables the continuous tracking of anvil clouds associated with detected deep convection after convective activity has stopped, enabling the study of the entire lifecycle of deep convective clouds and their associated anvils. Furthermore, we expect this framework to be applicable to a wide range of cases including the detection and tracking of low-level clouds and other atmospheric phenomena. In addition, this framework may be used to combine observations from multiple sources, including satellite observations, weather radar and reanalysis model data.

## 1 Introduction

Deep convective clouds (DCCs) are dynamical atmospheric phenomena resulting from instability in the troposphere. DCCs consist of a vertically growing core with a diameter of $10\,\mathrm{km}$ and updraught velocities of around $10\,\mathrm{ms^{-1}}$ (Weisman, 2015), and a surrounding anvil cloud formed due to horizontal divergence of cloud droplets lifted to the level of neutral buoyancy (Houze, 2014). The lifecycle of DCCs can be separated into three sections: a growing phase, where the core develops vertically; a mature phase in which the anvil cloud develops horizontally while convection continues within the core, and a dissipating phase in which the anvil cloud dissipates after convective activity ceases within the core (Wall et al., 2018). For isolated DCCs

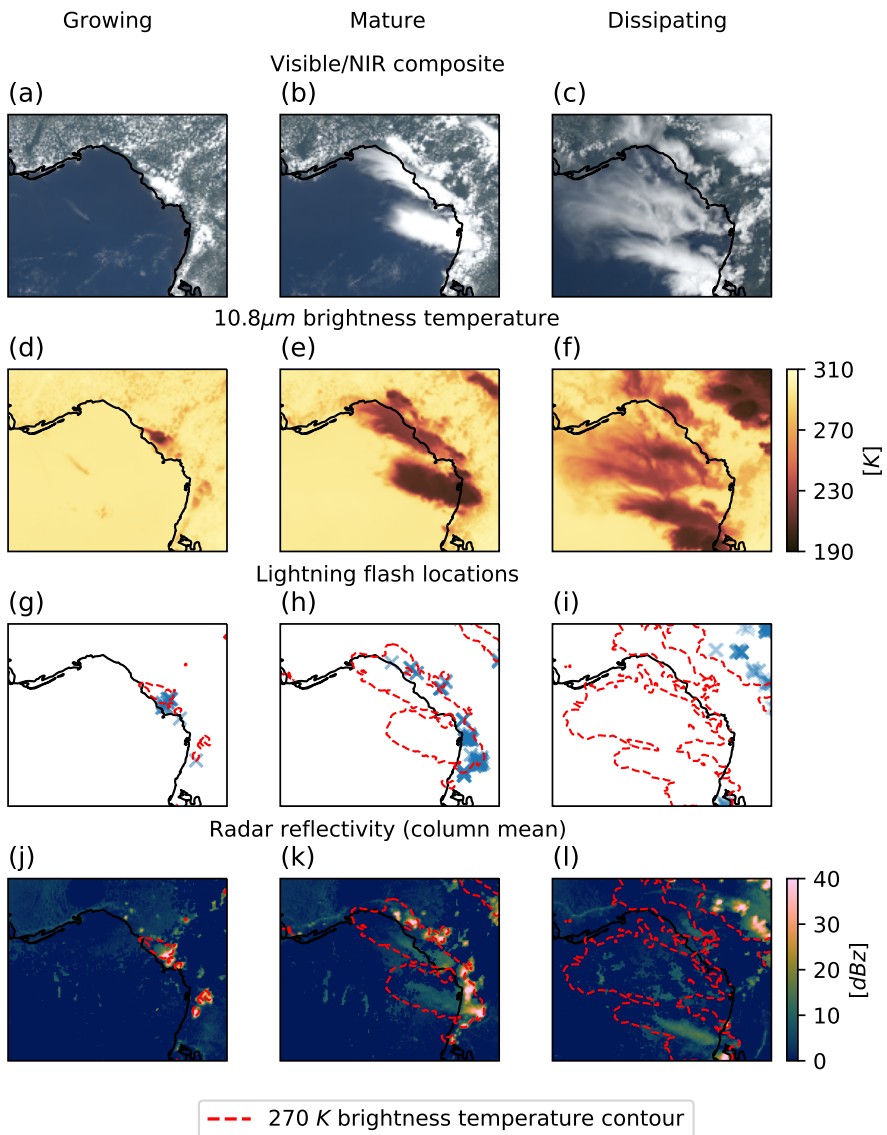

**Figure 1.** Observations of a cluster of deep convective clouds over North-West Florida throughout three stages of their lifecycle. This cluster of DCCs occurred on the afternoon of 19[th] June 2018. The "growing" column were observed at 17:00 UTC, the "mature" column at 19:00 UTC, and the dissipating column at 21:00 UTC. Note that, unless otherwise specified, this case study is used for all subsequent figures in this article.

– consisting of a single core – the overall lifecycle typically spans 1-3 hours (Chen and Houze, 1997). However, DCCs may also form with multiple cores feeding a single anvil cloud (Roca et al., 2017), and in these cases may span areas several orders of magnitude larger (Houze, 2004), and exist for 10-20 hours or longer (Chen and Houze, 1997).

DCCs are strongly linked with extreme weather events, including heavy precipitation, lightning and hail (Westra et al., 2014;
Houze, 2014; Williams et al., 1992; Bruning and MacGorman, 2013; Punge and Kunz, 2016; Matsudo and Salio, 2011). DCCs
are also strongly linked to global climate circulation and the global energy budget (Houze, 2004; Fritsch and Forbes, 2001;
Johnson and Mapes, 2001). Furthermore, the frequency and intensity of DCCs and their associated precipitation is expected to
increase with global warming, a prediction that is supported by both global climate model (Allen and Ingram, 2002; Trenberth
et al., 2003; Held and Soden, 2006; Muller and O'Gorman, 2011; O'Gorman et al., 2012; O'Gorman, 2015) and observational
evidence (Tan et al., 2015; Berg et al., 2013; Aumann et al., 2018; Houze et al., 2019). Improving our understanding of the
behaviour of DCCs and their interactions with the wider environment is vital for predicting the impacts of future climate change
(Westra et al., 2014).

Sequences of images from satellite instruments have been used to detect and track the motion of deep convective clouds
and tropical storms since the earliest geostationary weather satellites (Menzel, 2001). Whereas early detection and tracking
was performed by hand, numerous algorithms have been developed for the purpose of performing this task automatically, and
are widely used for both forecasting and research purposes (e.g. Mecikalski et al., 2011; Senf et al., 2015; Senf and Deneke,
2017; Feng et al., 2012, 2019; Zinner et al., 2008). There is a continual effort both to improve existing algorithms and develop
new methods to support these activities. However, it is important to understand the differences in observations of DCCs from
satellite imagery to those of other sources, particularly radar and lightning observations.

Visible and infra-red (IR) imagery from modern geostationary weather satellite instruments provide unique observations
of DCCs and their surrounding environment. Figure 1 compares observations of DCCs throughout three different stages of
their lifecycle between satellite visible and IR imagery, doppler cloud radar and lightning flash observations. Composite RGB
images from a combination visible and near-IR channels aboard the Advanced Baseline Imager (ABI) show a.: small, isolated
cores during the growing phase; b.: a large area of optically thick anvil during the mature phase, and c.: a large area of optically
thin anvil cloud during the dissipating phase. Brightness temperature imagery from the ABI 10.8 μm channel displays d.:
rapidly cooling cores; e.: a large, cold anvil cloud, and f.:, warmer brightness temperatures caused by thermal radiation from
the surface penetrating the optically thin dissipating anvil. Lightning flash locations observed by the Geostationary Lightning
Mapper (GLM) aboard GOES-16 shows g.: low frequency during the growing phase; h.: high frequency during the mature
phase, and i.: no lightning activity in the dissipating phase. Column mean radar reflectivity observed by NEXRAD doppler
cloud radar shows high radar reflectivity in the convective cores during the j. growing and k. mature phases, and no area of
high radar reflectivity during the dissipating phase. The outline of the region of brightness temperatures below 270 K observed
by ABI is shown by the orange dashed contour over the GLM flash locations and NEXRAD radar reflectivity to indicate their
observations relative to the anvil cloud.

These instruments are capable of observing the entire extent of the anvil clouds associated with DCCs over their entire
lifecycle, even after convective activity has ceased (fig. 1f). This is of particular importance due to the influence of anvil
cloud radiative forcing on the climate, their response to temperature change (Bony et al., 2016; Hartmann, 2016; Ceppi et al.,
2017; Gasparini et al., 2019) and possible feedbacks on subsequent convective activity (Varble, 2018). The newest generation
of geostationary imaging satellites offers greater opportunities for the study of DCCs due to their high spatial and temporal

resolution – allowing the detection and tracking of individual convective cores (Heikenfeld et al., 2019) – and also due to their
high signal to noise ratio allowing research quality observations (Iacovazzi and Wu, 2020).

The detection and tracking of DCCs from satellite imagery remains challenging due to the inability to directly observe the convection that drives DCCs using passive visible and IR observations. This is unlike radar and lightning observations, which can directly observe deep convection due to the strong correlations between core updraft intensity and radar reflectivity and polarisation (Austin, 1987; Rosenfeld et al., 1993; Zipser and Lutz, 1994), and lightning flash occurrence (Williams et al., 1989; Deierling and Petersen, 2008; Wang et al., 2017). Instead, a proxy for convective activity must be used to detect deep convection in visible/IR satellite imagery. The approaches used for this can generally be separated into two separate methods. Firstly, the use of thresholds on brightness temperature (BT) or other observed fields, which are capable of detecting DCC anvil clouds (e.g. Schmetz et al., 1997; Hong et al., 2005; Schröder et al., 2009; Liang et al., 2017; Senf et al., 2018). Secondly, the detection of rapidly growing cloud tops by observing changes in the anvil cloud top radiative cooling, or by other similar approximations of cloud growth (Zinner et al., 2008; Bedka et al., 2010; Müller et al., 2019).

Developing a detection method using either approach is made challenging by the dynamic nature of DCCs themselves. DCC cores typically have diameters of around 10km, and updraft velocities on the order of 10ms$^{-1}$ (Weisman, 2015), and exist for 1-3 hours (Chen and Houze, 1997). Large, mesoscale convective systems (consisting of multiple cores joined by a single large anvil (Roca et al., 2017)) may span areas several orders of magnitude larger than isolated DCCs (Houze, 2004), and typically exist to 10-20 hours or longer (Chen and Houze, 1997). The life cycle of a DCC can be split into three phases: an initiation or growing phase, a mature phase and a dissipating phase after the cessation of convective activity (Wall et al., 2018). There exists a significant difference between the diurnal cycles of deep convection over the land and over the ocean, with observed DCCs over land clustered towards the end of the day (Taylor et al., 2017).

The difficulties of detecting DCCs using various proxy approaches is demonstrated by the cross sections of an observed DCC over time in fig. 2. The observed brightness temperature of the DCC anvil cloud shows wide variation of over time, with the anvil cloud warming due to dissipation after the end of convective activity. This wide variety of observed temperatures leads to large differences in the chosen threshold value between different algorithm (see discussion in Bennartz and Schroeder, 2012). This choice of threshold value is further complicated due to the overlap in observed brightness temperatures between DCC anvils and non-convective clouds (Konduru et al., 2013). As a result, any detection method using a brightness temperature threshold must compromise between missed detection of DCCs, or false detections of non-DCC clouds.

The cooling of the cloud top is only visible for a short period during the initial phase of the DCC, before the anvil cloud top reaches the tropopause temperature after approximately 30 minutes. As a result, any method that solely relies on detecting the growth of the DCC will be unable to detect the anvil cloud after this initial growth phase has ended. While such algorithms provide an accurate detection of these early phases of DCC growth (Zinner et al., 2013), they are unable to continue tracking the anvil cloud after convective activity is no longer observed.

Fiolleau and Roca (2013) identified this need to compromise on the accuracy of detecting DCCs as a problem caused by the commonly used two-step framework for detecting and tracking DCCs. In this framework, DCCs are first detected in individual images, and then linked together over time in sequences of images. As a result, the detection method chosen must be

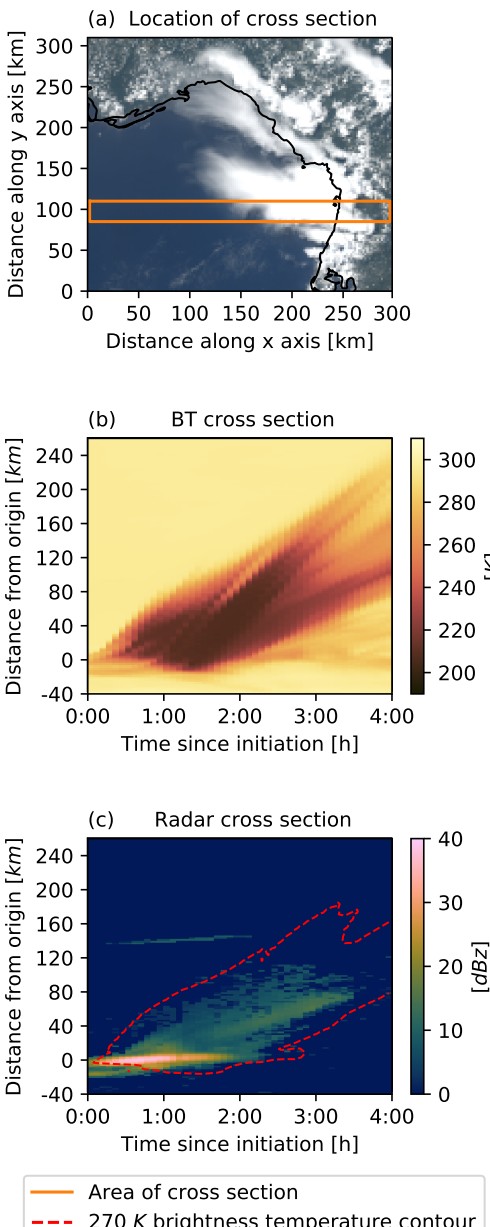

**Figure 2.** Cross sections of the DCC observed in figure 1 as they develop over time. a.: The location of the cross section within the observed DCC. The mean of values is taken in the North-South axis. b.: ABI 10.8 μm brightness temperature, showing rapid cooling for the first 30 minutes, followed by an expanding region of anvil cloud that begins to thin and warm after 2-3 hours. c.: Column mean radar reflectivity, showing the presence and location of the convective core. Initiation occurred at 82.0°W 28.5°N at a time of 17:00 UTC.

capable of detecting DCCs at each individual time step in order to track their entire lifecycle. Instead Fiolleau and Roca (2013) implemented a single-step framework for mesoscale convective systems that treats a sequence of images as a "3D" volume, and performs detection and tracking simultaneously by applying a watershed method over both spatial and temporal dimensions. Whereas this approach was successful for large, mesoscale systems, where the advection of the anvil is small compared to the overall anvil area, it is less capable of tracking small, rapidly moving convective cores. To improve the tracking of small DCCs, we have developed a semi-Lagrangian framework for single-step detection and tracking which accounts for the motion of DCCs using optical flow.

By utilising the semi-Lagrangian framework, we are able to combine the best elements of both growth-based and threshold-based detection methods. We show that it is possible to detect growing DCCs to a high degree of accuracy using methods similar to those of Zinner et al. (2008), and then extend the detected DCC over the entire anvil cloud using the "3D" watershed method of Fiolleau and Roca (2013). This framework reduces the compromise required between the rate of missed DCCs and falsely detected DCCs, improving the overall accuracy of our detection method compared to existing approaches. Furthermore, this method allows the anvil cloud to be detected and tracked even after the region of cloud top cooling is no longer detected. Finally, the "3D" method handles the merging and splitting of intersecting DCCs by detecting all DCCs that intersect at any point during their lifetime as a merged object.

## 2   Data

Three sources of data are used throughout this article. Primarily, visible and IR imagery from ABI aboard the Geostationary Operational Environmental Satellite (GOES)-16 weather satellite is used for the detection of DCCs. Secondarily, observations from the NEXRAD weather radar network and the geostationary lightning mapper (also aboard GOES-16) are used to assess and validate the tracking and detection method presented here.

### 2.1   Advanced Baseline Imager

The Advanced Baseline Imager (ABI) is a visible and IR radiometer aboard the Geostationary Operational Environmental Satellite (GOES)-16 series of weather satellites (Schmit et al., 2016). GOES-16, also known as GOES-East, is situated in a geostationary orbit at 75.2°W above the equator, providing a field of view (or "Earth-disc") covering most of the western hemisphere, including all of South America and most of North America. ABI has 16 channels operating in a range of spectral bands in the visible, Near-IR and thermal-IR. The majority of these channels have a resolution of 2 km at the sub-satellite point, although this reduces to approximately 3 km across most of the continental United States due to the satellite viewing angle. ABI operates in a flexible scan mode, imaging the continental US (CONUS) once every 5 minutes, the full disc every 10 minutes (15 minutes prior to April 2019), and two mesoscale regions of approximately 2500 by 2500 km every minute. Additionally, it is capable of scanning the full-disc every five minutes if no other scans are performed. This combination of high spatial and temporal resolution makes ABI suitable for detecting and tracking small and developing DCCs, as well as providing the spatial coverage to also track large mesoscale convective systems (Heikenfeld et al., 2019).

**Table 1.**

| Instrument | ABI (GOES-16) | SEVIRI (Meteosat-11) | IMAGER (GOES-14) |
|---|---|---|---|
| Temporal resolution | 5 minutes | 15 minutes | 30 minutes |
| Nadir spatial resolution | 2 km | 3 km | 4 km |
| Number of IR LW window channels | 3 | 2 | 1 |
| Number of IR WV channels | 3 | 2 | 1 |
| Noise equivalent temperature | 0.1K @ 300 K | 0.25 @ 300 K | 0.09 K @ 300 K |

Comparison of data from ABI to that from older geostationary instruments.

Compared to older geostationary instruments, ABI has higher spatial and temporal resolution, more channels in both the LW IR window spectrum and the LW IR water vapour (WV) spectrum, and low noise (table 1) (Iacovazzi and Wu, 2020). This, combined with many of the channels being derived from those aboard the Visible Infrared Imaging Radiometer Suite (VIIRS), make the data from ABI more suitable for research purposes than that from older instruments (Heidinger et al., 2020). A number of artifacts are known to occur in ABI imagery (Gunshor et al., 2020). Although the majority of these artifacts are removed using the data quality flag associated with the ABI data, we have found a number of cases in which bad detector stripes (described in section 3.2 of Gunshor et al. (2020)) are not flagged in the data, and so detection and tracking of DCCs has not been performed during the time periods these artifacts occurred.

In this paper we have used the ABI level 2 multi-channel cloud and moisture imagery product (MCMIP) which provides calibrated reflectances and brightness temperatures for all ABI channels on a common grid (Schmit and Gunshor, 2020), using the 5 minute frequency imagery provided over the CONUS region. The case study shown in the figures throughout this paper is for a subset of the CONUS scan region centred at 83.7° W, 29.2° N, over the time period of 18:00:00 to midnight UTC on the 19th June 2018. Validation was performed on a subset of the CONUS scan region from 114 to 76° W and 24 to 45° N over the entirety of 2018. All data has been sourced through the National Oceanic and Atmospheric Administration Big Data Program.

### 2.1.1 Selection of ABI Channels and Channel Combinations

In order to have equal performance during both day and nighttime, a selection of longwave IR ABI channels are used for the detection and tracking of DCCs (see fig. 3). These channels consist of the long wave LW clean and dirty window channels at 10.8 μm and 12.3 μm respectively, and the upper and lower troposphere water vapour channels at 6.2 μm and 7.3 μm respectively. Whereas the LW window IR brightness temperature is commonly used for the detection of DCCs using threshold-based methods, we have decided not to use it for this purpose in this method due to the wide range of brightness temperatures observed within anvil clouds, and the variance of anvil cloud temperature because of changes in tropopause temperature due to meteorology and latitude. However, the information contained within this field is used to for the optical flow calculation of the cloud motion field.

Two additional combinations of channels are used to detect areas of deep convective cloud anvil. The water vapour difference (WVD) combination (fig. 3b) of the upper troposphere WV channel minus the lower troposphere WV channel has been shown to provide a high detection rate for deep convective clouds (Müller et al., 2018, 2019). In clear sky or low cloud conditions, WVD shows the temperature difference between the upper and lower troposphere of generally around -20 K. However, in the presence of high, thick clouds the 6.2 µm has an additional contribution from stratospheric water vapour resulting in a warmer, and in extreme cases positive WVD value (Schmetz et al., 1997). Because both the WV channels are strongly absorbed by water vapour in the lower troposphere, the WVD field is not affected by surface and low altitude features and provides clear distinction between thick, high cloud and the background across a wide range of situations. Müller et al. (2019) found that a threshold of -5 °C gave a high detection rate of anvil clouds. Furthermore, as the WVD values are relative to the lower stratosphere temperatures, this field is much less affected by location and meteorology than the LW IR channels. However, the WVD is still prone to the false detections of non-convective clouds when using a thresholding method as it cannot directly distinguish between thick, high-altitude cloud that are associated with deep convection and those that are not.

The Split Window Difference (SWD), consisting of the clean IR window channel minus the dirty IR window channel (fig. 3c), aids in the detection and separation of optically thin anvil cloud (including cirrus outflow) from optically thick anvil due to the difference in ice particle emissivity between these two channels (Heidinger and Pavolonis, 2009). As a result, this combination displays warm temperatures of around 10 K for thin, ice clouds, near 0 K for thick clouds, and approximately 5 K for clear skies due to the contribution of boundary layer water vapour. The SWD is, however, also sensitive to low level clouds and low level water vapour concentrations, and so cannot be used alone to detect DCCs. It remains important to consider the SWD field due to the difficulty in separating anvil clouds from cirrus when using LW IR BT alone (Hong et al., 2005). By subtracting the SWD from the WVD field, we can reduce the sensitivity of our detection scheme to cirrus clouds, reducing the rate of erroneous detections. Further, adding the SWD field to WVD field can enhance the appearance of cirrus, enabling the detection of thin ice cloud associated with cirrus outflow and dissipating anvils.

## 2.2 Geostationary Lightning Mapper

The Geostationary Lightning Mapper (GLM) is also mounted on GOES-16 and detects lightning flashes using an optical transient detector. The optical transient detector utilises a single, narrow-band near-IR channel centred on 777nm (Orville and Henderson, 1984) to detect momentary changes in brightness associated with lightning events at a frequency of 400µs (Christian et al., 2003), providing a 70% minimum efficiency of detection (Goodman et al., 2013). GLM has the same field of view as the ABI instrument, albeit with a lower spatial resolution of 8 km at the sub-satellite point.

As lightning observations are strongly correlated with DCCs, data from GLM is used to validate the detection of DCCs using ABI. The level 2 GLM Lightning Cluster-Filter Algorithm product provides a dataset of events, groups and flashes processed from the GLM data (Peterson, 2019), and filters artifacts from the level 1 GLM data (Peterson, 2020). From this dataset we extract detected flashes as evidence of DCC occurence. These locations are then processed by mapping their frequency onto the ABI grid for validation of the algorithm.

## 3   Method

We present here a novel method for detecting and tracking both the growing cores and anvils clouds of DCCs, consisting of the following steps:

1. Ingest of LW IR BT fields from geostationary satellite imagery, including calculation of WVD and SWD fields from IR water vapour and LW IR window channels.

2. Calculation of optical flow vectors to be used as an *a priori* estimate of cloud motion for use in the semi-lagrangian framework.

3. Detection of growing DCC cores using cloud top cooling rate.

4. Detection of thick and thin anvil clouds associated with detected cores using a semi-lagrangian "3D" watershedding method.

5. Grouping of cores into multi-core systems, calculation of statistics and validation using lightning observations.

### 3.1   Estimation of cloud motion vectors using optical flow

The retrieval of atmospheric motion vectors (AMVs) has been performed since the earliest geostationary satellite observa-
tions (Menzel, 2001). AMVs provide information about the motion of clouds in the atmosphere, including DCCs (Bedka and Mecikalski, 2005), and are routinely generated for the majority of operational geostationary earth observation satellites, including GOES-16 (Daniels et al., 2016). However, although AMVs may provide useful information about the motion of DCCs, the non-geostrophic nature of wind fields in these conditions may result in the AMVs being calculated inaccurately or rejected by quality control checks (Bedka and Mecikalski, 2005).

Optical flow algorithms are a family of algorithms used to estimate the apparent motion of objects observed in a series of images (Aggarwal and Nandhakumar, 1988). A wide range of optical flow algorithms exist, and these have been successfully applied to many computer vision applications. It should be noted that optical flow does not necessarily represent the physical motion of an object, and is instead an estimation of the relative motion between an object and the observer and additionally any change in the apparent object (including growing, shrinking or other warping of the object).

Optical flow algorithms have been previously shown to be accurate for the prediction of AMVs using geostationary satellite images (Wu et al., 2016), as long as the observations are sufficiently frequent such that the motion of unique features between images is less that the length scale at which neighbouring features can be resolved (Bresky and Daniels, 2006). Heikenfeld et al. (2019) found that at imaging frequencies of less than 5 minutes the motion of DCC cores was less than the spacing between neighbouring cores in the majority of cases, indicating that the frequency of the ABI CONUS scan region is suitable
for calculating optical flow vectors of DCCs. The use of optical flow has several advantages over traditional AMVs for the retrieval of DCC motion vectors: optical flow can be calculated quickly using only two subsequent images and no *a priori* information, aiding in near real-time applications; and also have no requirement for geostrophic balance. Optical flow algorithms

are routinely used in the nowcasting of convective precipitation, and can be used to provide accurate predictions of DCC with an hour of lead time using either radar or satellite observations (e.g., Bowler et al., 2004; Bechini and Chandrasekar, 2017; Woo and Wong, 2017).

It should be noted that we are using optical flow to estimate the apparent motion of the cloud field between subsequent images, with the aim of using these vectors to map the locations of DCCs from one step to the next, instead of calculating actual AMVs corresponding to winds. This approach avoids a number of challenges with the use of optical flow for calculating AMVs including the estimation of the height of estimated flow vectors and the detection of diverging or converging motion vector fields in situations of growing and dissipating clouds respectively. In the latter case, we in fact aim to include the divergence and convergence within the optical flow vector field to map both the location and shape of observed clouds between time steps.

We use the Farnebäck algorithm (Farnebäck, 2003) to estimate optical flow vectors of observed DCCs. The Farnebäck algorithm calculates a 'dense' field of optical flow vectors, in which a flow vector is calculated for every pixel in the origin image which maps to its predicted location in the destination image. This calculation is performed by finding the minimum cross-correlation over increasingly smaller subsets of the image. This iterative approach allows flow vectors to be calculated to sub-pixel accuracy. In this framework, we have used the implementation of the Farnebäck algorithm from the OpenCV image processing package (Bradski, 2000). Although other optical flow algorithms may provide better accuracy in different circumstances (Baker et al., 2011), we have found that the ability of the Farnebäck algorithm to accept a range of parameters is important for detecting the motion of clouds across a wide range of scales. The choice of parameters used is a compromise between the ability to robustly detect flow vectors in areas of the image with low contrast between features (e.g. in the centre of anvil clouds), versus the fidelity of the motion vectors detected for small, high contrast features such as developing cores. Table 2 shows the parameters chosen for the Farnebäck algorithm for ABI imagery in the CONUS scan region, which has a temporal resolution of 5 minutes and the spatial resolution of 2 km. The values used for the window size parameter is scale dependent, proportional to the time between subsequent images and inversely proportional the spatial resolution.

An example of the motion vectors calculated by the Farnebäck algorithm when applied to the 10.8 μm brightness temperature field is shown in fig. 4. By comparing the predicting flow vectors to the future evolution of the cloud field (dashed line), we can see that the algorithm correctly estimates the future evolution of the anvil cloud. Optical flow, and similar motion vector techniques, have been successfully applied to both the detection of developing deep convection (Zinner et al., 2008; Zhang et al., 2014) and tracking detected deep convective features (Senf et al., 2018) separately.

We estimate the uncertainty of the optical flow vectors by comparing the residual error in the location of features in subsequent images after the detected motion from optical flow is accounted for (see fig. 4d). We restrict the estimation of uncertainty to regions of clouds with brightness temperatures less than 270 K as these are the situations in which we see enough contrast with the background to detect optical flow motion vectors. We find that in the majority of cases that the relative uncertainty in the magnitude of the location offset versus the magnitude of the estimated optical flow vector is less than 10 %, with mean and median relative uncertainties of 15.0 % and 8.4 % respectively.

**Table 2.**

| Parameter | Value |
| --- | --- |
| Window size | 16 |
| Levels | 5 |
| Pyramid scale | 0.5 |
| Iterations | 3 |
| Polynomial order | 5 |
| Polynomial sigma | 1.1 |
| Window type | Gaussian |

The values of parameters used in the
Farnebäck algorithm for detection of
optical flow vectors in this application.

Two potential sources of uncertainty are the assumptions made by the Farnebäck algorithm that the feature being tracked remains the same size and intensity in subsequent images. For optical flow tracking using brightness temperature images of growing DCCs, neither of these assumptions are true. However, we have taken steps to reduce the impact of both these sources of uncertainty on the tracking algorithm. For the first of these assumptions, we find that in the case of small, fast moving DCCs – where the accuracy of the optical flow vectors is most important – the changes in size of the DCC is small compared to the overall motion, and so the uncertainty introduced is small. Comparably, for large DCCs where the changes in size may be large compared to the motion, the uncertainty introduces has less impact on the tracking algorithm. In the worst case scenario the estimated optical flow field will be zero, in which case the "3D" detection and tracking algorithm works in the same manner as that of Fiolleau and Roca (2013), which is suitable for use on these larger DCCs. To reduce uncertainty caused by the second assumption we normalise the range of brightness temperatures between subsequent images when estimating optical flow to reduce the change in brightness temperature observed for growing DCCs.

### 3.2 A Semi-Lagrangian Framework for Morphological Image Processing

Morphological image operations analyse images using their geometrical and structural properties. Core to many morphological algorithms, from simple filters to complex neural networks (Kalchbrenner et al., 2014), is the kernel, or convolution method. A convolution method performs operations on the pixels of an image by applying a convolution stencil to the pixel and its neighbours. In a conventional convolution scheme, such as that used in the methods of Fiolleau and Roca (2013), the convolution stencil acts on adjacent pixels in both time and space (see fig. 5a). In this Eulerian framework, different locations in time are considered in the same manner as those in the spatial dimensions. However, we know from previous analysis of DCCs that the motion of convective cores between images can be similar to the spacing of cores and their size (Heikenfeld et al., 2019). As a result, it is important to include the effects of advection when comparing images across time steps.

To perform morphological operations which take into account this advection, we have developed a novel Lagrangian convolution method. For spatial operations, the Lagrangian stencil operates identically to that of a classical convolution method. However, when sampling points at prior or subsequent time steps, the location of the stencil are offset by the relevant optical flow vectors (fig. 5b). Values at the offset stencil locations are interpolated, providing a Lagrangian reference frame for changes in the observations over time. When applying the convolution stencil to every pixel in a sequence of images, this provides a semi-Lagrangian framework for morphological operations, combining the Lagrangian reference frame for evaluating changes over time while maintaining the regular grid of the images.

We have developed new implementations of several existing image processing operations within the Lagrangian convolution framework, including:

- Sobel edge detection (Sobel, 2014)

- Watershed segmentation using the connected-components method (Bieniek and Moga, 2000)

- Labelling of connected components (Hoshen and Kopelman, 1976)

These operations are used in this method to detect the full extent of the anvil cloud associated with the DCC, to perform detection continuously across multiple time periods while accounting for the motion of the DCC, and to identify individual DCCs and DCC clusters across multiple time periods respectively.

The Sobel method detects edges in an image using the magnitude of the local gradient at each pixel. Edge detection enables the segmentation of an image into separate regions without pre-defined thresholds (such as in brightness temperature) to separate them.

Watershed algorithms are a method of image segmentation that equate an image to a topographical map, with elevation according to the value of the pixel. Each pixel is then descended towards its local minima until it reaches a predefined marker region. The method takes its name from the geographical feature of the same name, which refers to the separation between adjacent drainage basins. Although this physical interpretation of the algorithm applies to two dimensional images, the method can be applied to arrays with any number of dimensions, such as the method used by Fiolleau and Roca (2013) which applied watershedding to a three dimensional field.

Labelling algorithms assign unique identifiers to each segmented region provided by either the edge detection or watershed algorithms.

### 3.3 Detection of Growing Deep Convection

Growing deep convective cores are detected in a similar manner to that used by Zinner et al. (2008). We have found that the WVD field provides the best observations for detecting growing deep convective cores as the field isolates growth in the mid troposphere, removing spurious observations of growth due to boundary layer convection and cloud formation. The growth is calculated using the finite difference of the WVD field in the Lagrangian perspective.

We classify a region of growing deep convection as a region of continuous warming of the WVD field of at least 0.5 K per minute over a 15 minute period, covering an area of at least 3 by 3 pixels (approximately 9 by 9 km) at each time step. This

threshold for cloud top growth is based on previous studies which have found a cooling of 8K over a 15 minute period to be a good predictor of intense convection (Roberts and Rutledge, 2003; Hartung et al., 2013). The region of growing cloud is then expanded through the use of a watershed operation to fill surrounding areas of the cloud field with a detected growth rate greater than 0.25 K per minute to detect weaker areas of updraft within the growing cores. Finally, each region of detected growth is labelled, and each label checked to ensure that the growth region ends with a WVD field with a value of greater than

310 -5, indicating the formation of an anvil cloud (Müller et al., 2018).

Figure 6 shows a comparison between the detected core cooling rates in ABI imagery and the corresponding column radar reflectivity measured by NEXRAD. During the early development of the core, the detected cooling rate (fig. 6a) shows growth in the same locations as the NEXRAD radar reflectivity (fig. 6c). However, during the mature stage of the DCC, discrepancies develop between the observed cooling rate (fig.6b) and the radar reflectivity (fig. 6d) due to the development of the anvil cloud

blocking satellite observations of the core underneath.

### 3.4 Detection of Anvil Clouds

The region of anvil cloud associated with the growing convective clouds detected in the previous section is detected and tracked using an edge-base watershed segmentation approach. The edge-based approach to cloud detection avoids the use of a fixed threshold for anvil temperature, and so can detect a more accurate extent of the anvil cloud (Dim and Takamura, 2013). We

define an upper threshold for the WVD field of -5 K, as used by Müller et al. (2018), and a lower threshold of -15 K, which we define as definite non-anvil cloud. Because the presence of thin cirrus outflow from the anvil clouds can make it difficult to determine the extent of the anvil cloud, we use the SWD field as described in section 2.1 to either remove or include the region of cirrus outflow in the detected anvil region. To detect the thick anvil cloud, we subtract the SWD field (fig. 7a). In this case, the upper and lower threshold remain the same as the SWD field is approximately 0 K for thick, high clouds, and so

has no effect on the temperature of these features. For detecting the thin anvil region, we add the SWD field and increase the value of both thresholds by 5 K to 0 K and -10 K respectively (fig. 7c). This change is made to account for the effect of low level water vapour on the SWD field which gives a background value of approximately 5 K. Between these two thresholds, we have a region in which we are uncertain of the extent of the anvil cloud. By applying a Sobel filter to detect the local gradient magnitude of the combined WVD/SWD field (Sobel, 2014), we detect the outer extent of the anvil cloud within this region

where we see the greatest magnitude in the detected edges (see fig. 7b,d).

When applied to the detected edges of the anvil clouds using the sobel filter, with the growth regions detected previously as markers, the watershed method allows us to detect those anvil regions associated with detected regions of growing DCCs, while avoiding the detection of non-convective regions of high, cold cloud. Furthermore, due to the application of the watershed algorithm to both the spatial and temporal dimensions of the sequence of images through the semi-Lagrangian framework, we

are able to detect the associated anvil clouds after the growth of the DCC is no longer observed (see fig. 8).

Figure 8 shows an example of the results of detecting and tracking DCC cores and their associated anvils. Detection of the cores (outlined in red) and the initial development of the associated anvils (outlined in orange and blue for the thick and thin anvil regions respectively) can be seen in fig. 8a. In fig. 8b we see the development of the mature anvil, which primarily consists

of thick anvil, and secondary core detections as new convection develops at the edge of the DCC. In fig. 8c, we see the detected anvil cloud beginning to dissipate, and a larger proportion of the anvil cloud detected as thin anvil. At this point in the lifetime of the tracked DCC we no longer observe any growing core, however the "3D" approach allows the continued tracking of the anvil cloud until it dissipates.

## 4 Evaluation

The effectiveness of the semi-Lagrangian framework for the detection of DCCs is evaluated by analysing the proximity of detected anvil cloud regions to lightning flash detection from GLM. Lightning observations are frequently used to validate detection methods for deep convection (e.g., Zinner et al., 2013; Müller et al., 2019) due to the strong correlation between deep convective updraughts and lightning activity. Although GLM is not capable of detecting all lightning events (approximately 70% of lightning events are detected) (Peterson, 2020), the high frequency of lightning flashes per DCC mean that these observations provide a suitable ground truth for validation. It should be noted that lightning observations are only suitable for validating the detection of the thick anvil region, as lightning does not occur in the cirrus outflow. As a result, validation of the detection of the thin anvil region would require the use of other data such as cloud profiling radar or lidar observations, and is not considered further in this paper.

Here we apply the same validation method as used by Müller et al. (2019) to evaluate the semi-Lagrangian framework for the detection of DCCs. We classify detection events into three categories:

– Correct detection (CD), when the algorithm detects a DCC that is collocated with one or more lightning observations

– False detection (FD), when the algorithm detects a DCC but no lightning flash is observed

– Missed detection (ND), when the algorithm does not detect a DCC but a lightning flash is observed

Using these three categories of events we can define two measures of accuracy for the detection of DCCs. The probability of detection (POD) is defined as the number of correct detections divided by the total number of correct and missed detections. This provides a measure of how likely the algorithm is to detect a DCC that exists in the ground truth. The false alarm rate (FAR) is defined as the number of false detections divided by the total number of correct and false detections. This provides a measure of how likely a DCC detected by the algorithm is not present in the ground truth.

When evaluating whether detected DCC regions and lightning observations were collocated Müller et al. (2019) considered events within 32 km and 15 minutes to be collocated. This margin of uncertainty was separated into two section, half from the physical separation between observed lightning strikes, and the remaining half from uncertainty in the collocation and geolocation of the satellite and lightning observations. For a typical ABI pixel length over the continental United States of 3 km, this margin of error translates into 10 pixels in the ABI view. The distance between a GLM lightning flash and detected cloud region is defined as the distance between the flash and the nearest ABI pixel within that region, with GLM flashes that fall within a detected DCC given a distance of 0. When considering that the resolution of GLM is a factor of four less than

**Table 3.**

| Detection Method | n | POD | FAR |
|---|---|---|---|
| Growth based | 70,527 | 0.23 | 0.27 |
| WVD threshold | 312,591 | 0.99 | 0.73 |
| Semi-lagrangian | 14,479 | 0.93 | 0.35 |

Probability of detection (POD) and false alarm rates (FAR) for three different detection methods validated against observed GLM flashes (n=47,096,707). Growth based refers to the detection of growing DCCs using the method described in section 3.3. WVD threshold uses the threshold method developed by Müller et al. (2018). Semi-Lagrangian refers to the detection of anvils clouds connected to growing cores using the edge-based watershedding method described in section 3.4.

that of ABI, we consider that the same justification for the margin of error used by Müller et al. (2019) is also applicable to collocated observations from ABI and GLM.

     Validation was performed using GOES-16 ABI data from the CONUS scan region for the entirety of 2018, which was processed using the method described in this article. In total validation was performed for 290 days of ABI data, the remaining 75 days being excluded due to missing observations from either the ABI or GLM instruments aboard GOES-16, or artifacts 375     present in the ABI data. Detection and tracking of DCCs was performed on a subset of the CONUS scan region for each 25 hour period consisting of the array locations of 500-1750 in the x dimension and 250-1000 in the y dimension, corresponding to a bounding box of 113.6 °W to 76.2 °W and 24.5 °N to 44.2 °N respectively. By performing validation over both a large region, including a range of both land and ocean domains, and a full year time period, we aim to avoid any bias in the validation associated with the variability of the accuracy of the method with location and season.

Results of the validation of the detected anvil region, as well as those for the detection of growing deep convection and the WVD filter are shown in table 3. The regions of growing DCCs detected using the method described in section 3.3 shows low scores for both the FAR and POD metrics. While the detection of growing DCCs shows a low FAR of 0.27, the short time frame in which growth can be observed leads to a high rate of missed detections of lightning flashes, which results in a POD of 0.23.

For comparison, we also evaluate the accuracy of detecting anvils only by a fixed threshold of the WVD without detecting growing cores, as used by Müller et al. (2018). Compared to the detection of growing DCC regions, the WVD threshold shows a much higher POD of 0.99, but also has a high FAR of 0.73, repeating the findings of Müller et al. (2019) which show that although the WVD threshold method is capable of detecting the majority of DCCs, in is incapable of distinguishing between anvil clouds and other thick, high altitude clouds. Furthermore, the WVD threshold detection detects a much larger number of 390     clouds (n=312,591) compared to either of the other detection methods, further indicating that a large number of non-convective clouds are detected using the threshold method on its own.

Finally, the anvil regions detected using a combination of the detected growth regions and the WVD field using the semi-Lagrangian framework described in section 3.4 is validated. The novel method has a high POD of 0.93 similar to that of the WVD threshold, while also maintaining much of the low FAR of the detection of growing DCCs (FAR=0.35). This result highlights the capability of the semi-Lagrangian detection framework to use growth-based detection methods to substantially reduce the compromise between POD and FAR error rates by combining multiple methods for the detection of DCCs.

## 5 Conclusions

Algorithms for the detection and tracking of deep convective clouds perform a vital role in both forecasting and research applications. Sequences from geostationary satellites provide unique observations of DCC anvil clouds over their entire lifecycle. However, the traditional framework used by such algorithms requires a compromise between the rates of false and missed detections due to the overlap in signature from convective and non-convective clouds (Konduru et al., 2013). Whereas novel methods have approached this problem for the detection of large, mesoscale convective systems (Fiolleau and Roca, 2013), such approaches do not take advantage of the capability of the latest generation of geostationary imaging satellites to detect individual deep convective cores.

By developing and implementing a novel semi-Lagrangian framework for the detection and tracking of DCCs we are able to combine the detection of growing DCC cores (Zinner et al., 2008) and DCC anvils (Müller et al., 2018) to detect and track DCCs over their entire lifecycles. The novel methods developed here for the Semi-Lagrangian computer vision framework, along with implementations of multiple image processing operations commonly used for object detection, allow the accurate detection and tracking of moving objects utilising both spatial and temporal information. These methods may have impacts on applications of computer vision beyond the detection and tracking of DCCs. Furthermore, the novel framework is able to achieve higher levels of accuracy without compromising on the number of DCCs detected, as with previous algorithms (Müller et al., 2019).

By using this novel methodology, we are able to detect and track both small, isolated DCCs and large, mesoscale convective systems with a high degree of accuracy, high spatial and temporal resolution and across large domains such as the continental United States. The data provided about the behaviour of DCCs over their entire lifetime will allow new research into vital topics such as the response of deep convection and climate change, and the interactions and feedbacks between DCCs and large scale atmospheric thermodynamics (Varble, 2018).

*Code and data availability.* The methods described in this paper are made available through a python module released under the BSD 3-clause licence. The python module can be accessed through the following github repository: https://github.com/w-k-jones/tobac-flow. The version of the code used to for this paper, including both the generation of figures and the validation of the method can be accessed through the following release: https://github.com/w-k-jones/tobac-flow/releases/tag/v1.0 (Jones, 2022b). The figures produced for this article can be reproduced through the jupyter notebook included in the repository: https://github.com/w-k-jones/tobac-flow/blob/master/examples/Tracking%20Paper%20Plots.ipynb.

All ABI, GLM and NEXRAD data used in this paper is openly available through the NOAA big data program. The results of the validation
described in section 4 can be obtained from the following data record: https://zenodo.org/record/5885722 (Jones, 2022a).

*Author contributions.* WKJ lead the development of the detection and tracking framework, data analysis and validation, and wrote this paper
with contributions from MC and PS.

*Competing interests.* The authors declare that they have no competing interests.

*Acknowledgements.* This research was supported by the European Research Council (ERC) project constRaining the EffeCts of Aerosols
on Precipitation (RECAP) project under the European Union's Horizon 2020 research innovation programme with grant agreement number
724602.

This work was performed on JASMIN, the UK collaborative data analysis facility, and LOTUS, the associated high performance batch
compute cluster.

The authors would like to thank the NOAA big data program for making the data used in this paper openly available.

Thanks go to Max Heikenfeld and Fabian Senf for many fruitful discussions on the tracking of deep convective clouds, and for the
development of the tobac software package.

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

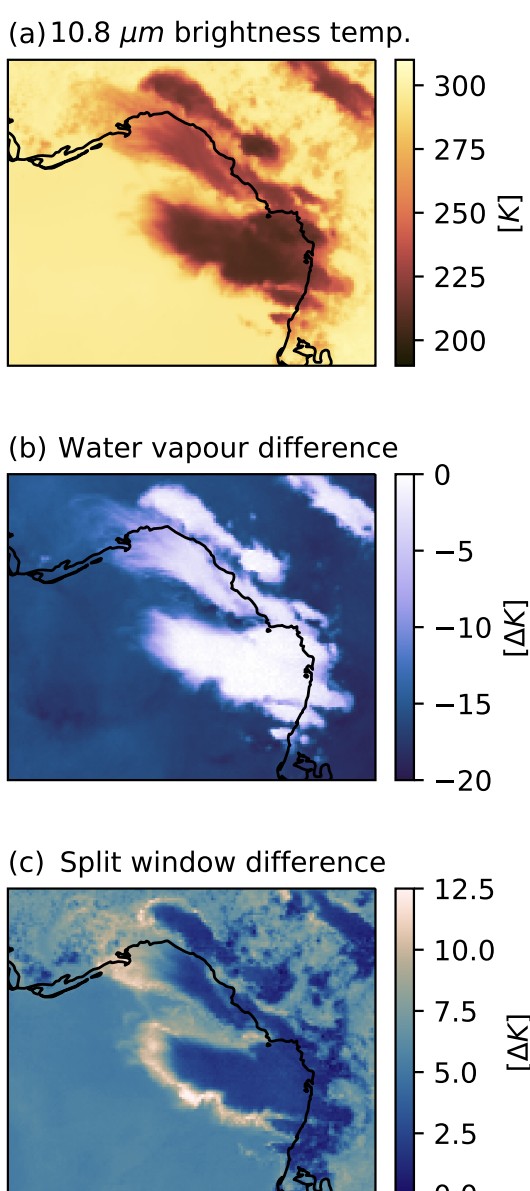

**Figure 3.** ABI channels and channel differences used with the detection and tracking algorithm. a.: The 10.8 μm brightness temperature, or clean longwave window channel, which can differentiate clouds at all altitudes by their brightness temperature. b.: the water vapour difference (WVD) combination, of the 6.2 μm upper troposphere water vapour channel minus the 7.3 μm lower troposphere water vapour channel, which is strongly negative for clear sky and low cloud, but approaches positive values for thick, high clouds. c.: the split window difference (SWD) combination of the 10.8 μm clean longwave window channel minus the 12.3 μm dirty longwave window channel, which is near zero for thick clouds, around 5 K for clear skies and approximately 10 K for thin, ice clouds.

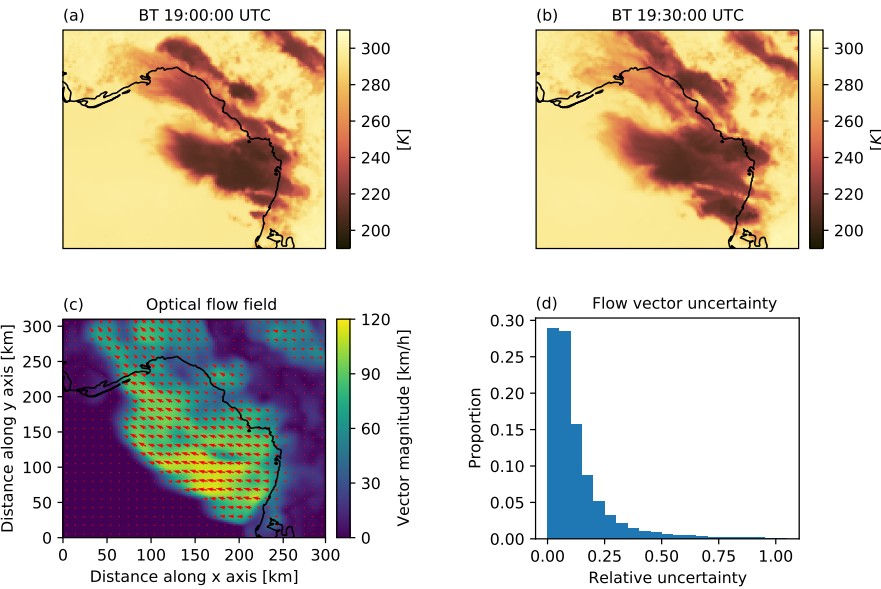

**Figure 4.** Cloud motion field for the DCC observed in figure 1 calculated using optical flow over a 30 minute period. a.: The initial image and b.: the final image. c.: The calculated optical flow vector field, with the detected motion displayed by the red arrows and the velocity of the vectors by the background. Optical flow vectors are estimated across the entire domain, but are here only visible for the region of high anvil cloud. d.: the relative uncertainty distribution of calculated optical flow vectors

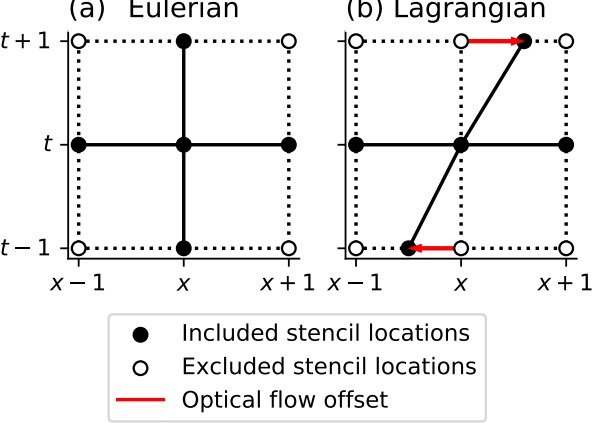

**Figure 5.** A comparison of convolution stencils with square connectivity in Eulerian (a) and Lagrangian (b) frameworks. In the Lagrangian framework, the points at prior and subsequent time steps are offset by the calculated optical flow field.

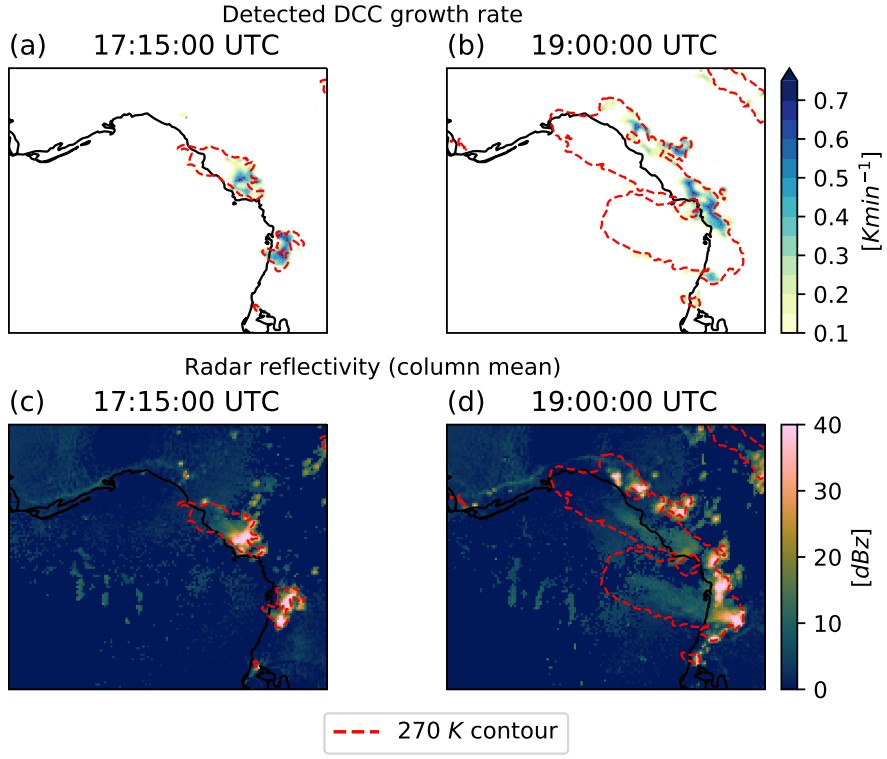

**Figure 6.** Detection of growing deep convective cloud regions for the DCC cluster in figure 1. 15 minute average cooling rate in the GOES-16 ABI water vapour difference field is used to detect growing cores. This is effective in the growing phases of convection (a.), but becomes less effective during the mature phase (b.). Comparison with NEXRAD column mean radar reflectivity remapped to the ABI grid(c., d.). An average cooling rate of greater than $0.5$ Kmin$^{-1}$ is indicative of a growing convective core.

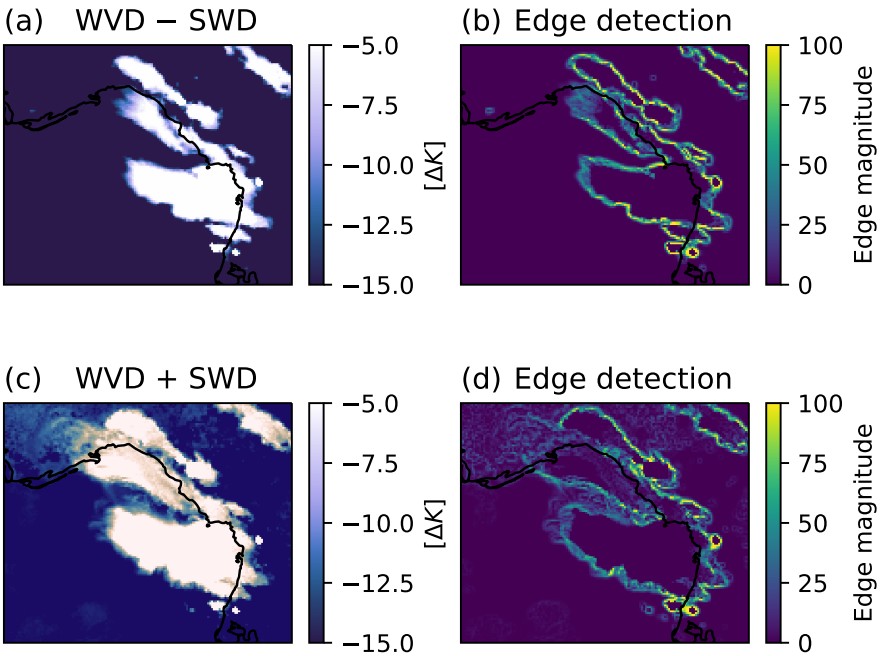

**Figure 7.** Detection of anvil cloud extent for the mature DCC cluster in 1 using the edge gradient method. a.: The combined field of the WVD minus the SWD, to isolate the thick anvil, between the upper and lower thresholds of -5 and -15K respectively. b.: the detected edge gradient magnitude of the field between these thresholds, which is used to detect the outer extend of the thick anvil cloud. c.: the combined field of WVD plus the SWD, to enhance the thin anvil, and d.: the calculated edge magnitude of this field.

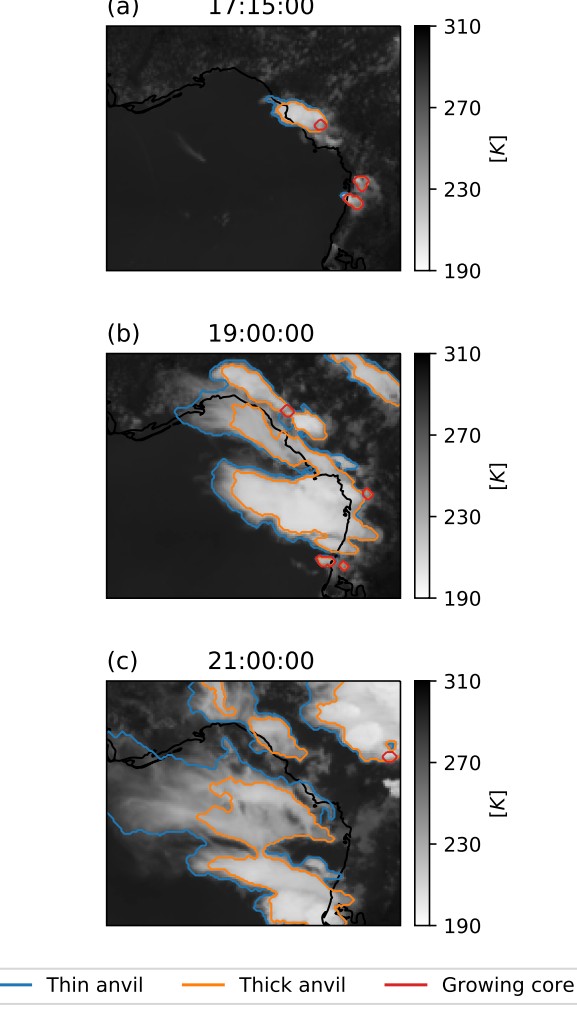

**Figure 8.** Detected regions of thin anvil cloud (blue), thick anvil cloud (orange), and developing cores (red) overlaid on the GOES-16 ABI 10.8 μm brightness temperature field for the DCC cluster from figure 1. The three stages of the DCC lifecycle are shown; the growth phase (a.), the mature phase (b.), and the dissipating phase (c.). Note that the anvil region continues to be detected in c. after growing cores are no longer detected.