# Peer review of "A Semi-Lagrangian Method for Detecting and Tracking Deep Convective Clouds in Geostationary Satellite Observations"

_Atmospheric Measurement Techniques, 2022_

## Author Response (AR1)

In addition to the changes made below in response to reviewers, we have made changes to all the plots in order to ensure that they are clear and accessible to all readers.

**Reviewer 1**

This paper proposes a new algorithm for the detection and tracking of DCC from geostationary observations. The data and method are very well presented, and the manuscript is well written and easy to read. This new method addresses some issues related to DDCs detection at any moment of their lifetime and their movement tracking, showing good performances. The method claims to improve the accuracy of the current detection algorithms; however, I believe it would benefit the paper to place this work in a broader context. It could be beneficial to the paper to mention the accuracy of some of the most common DCCs detection algorithms for missed and falsely detected DDCs. In general, the manuscript lacks some data to support and justify certain affirmations. In the minor comments I added some parts that could benefit from a better context, in my opinion.

Direct comparison with the accuracy of existing detection algorithms is difficult due to the lack of an agreed standard for how to validate the accuracy of different algorithms. We agree however that more context is required here, so we have further elaborated on the challenges faced by existing algorithms and how our new algorithm addresses these.

**Minor comments:**

**1. In the introduction, please briefly introduce the concept of DDCs.**

We have included a new paragraph on DCCs, focusing on the properties of importance to detection and tracking, starting at L20 in the updated manuscript:

Line 20: "Deep convective clouds (DCCs) are dynamical atmospheric phenomena resulting from instability in the troposphere. DCCs consist of a vertically growing core with a diameter of 10 km and updraught velocities of around 10 ms-1 (Weisman, 2015), and a surrounding anvil cloud formed due to horizontal divergence of cloud droplets lifted to the level of neutral buoyancy (Houze, 2014). The lifecycle of DCCs can be separated into three sections: a growing phase, where the core develops vertically; a mature phase in which the anvil cloud develops horizontally while convection continues within the core, and a dissipating phase in which the anvil cloud dissipates after convective activity ceases within the core (Wall et al., 2018). For isolated DCCs, – consisting of a single core – the overall lifecycle typically spans 1-3 hours (Chen and Houze, 1997). However, DCCs may also form with multiple cores feeding a single anvil cloud (Roca et al., 2017), and in these cases may span areas several orders of magnitude larger (Houze, 2004), and exist for 10-20 hours or longer (Chen and Houze, 1997).

This has replaced the paragraph at L66, which has now been removed.

2. All the references to figures throughout the text should be if within the text or Figure if at the beginning of a sentence.

We have edited the text to correct this.

3. L52-55 Maybe it would be useful to give some numbers here (e.g., the spatial resolution improved from ~5-7km to ~1km) to give an order of magnitude of the improvements with the newest generation.

We have moved the discussion of ABI capabilities vs older sensors to the data section, and added a table comparing the properties of the ABI instrument aboard GOES-16 to previous instruments to show the improvements in resolution, channels and SNR.

4. Acronyms need to be introduced only once. Please double check your manuscript for this (e.g. L107). On the other hand, some acronyms were not introduced (e.g. L130 LW)

We have checked and corrected the introduction of acronyms throughout the text.

5. *L121-122 Please add again some numbers here for SNR to give an order of magnitude of the improvements.*

This has been included in the table referred to in response to point 3, with a comparison both SNR of the IR channels and an estimation of the SNR of the water vapour difference combination. We have added the following:

Line 131: "Compared to older geostationary instruments, ABI has higher spatial and temporal resolution, more channels in both the LW IR window spectrum and the LW IR water vapour (WV) spectrum, and low noise (table 1) (Iacovazzi and Wu, 2020)"

**6. L140-141 This sentence is not clear, please reformulate**

The sentence was missing a word and has now been corrected to:

"However, in the presence of high, thick clouds the  $6.2 \mu m$  channel has an additional contribution from stratospheric water vapour resulting in a warmer, and in extreme cases positive, WVD value (Schmetz et al., 1997)".

**7. L176 What does sufficiently mean? Please give more data.**

In this case, "sufficiently" means in relation to the motion of observed DCCs between subsequent images in comparison to the spacing of those features. We have included an additional reference to show what the motion of detected DCCs is in relation to the spacing between DCCs to demonstrate how the improved frequency of imagery from GOES-16 ABI allows individual DCCs to be identified between time steps, as follows:

Line 210: "Optical flow algorithms have been previously shown to be accurate for the prediction of AMVs using geostationary satellite images (Wu et al., 2016), as long as the observations are sufficiently frequent such that the motion of unique features between images is less that the length scale at which neighbouring features can be resolved (Bresky and Daniels, 2006). Heikenfeld et al. (2019) found that at imaging frequencies of less than 5 minutes the motion of DCC cores was less than the spacing between neighbouring cores in the majority of cases, indicating that the frequency of the ABI CONUS scan region is suitable for calculating optical flow vectors of DCCs."

**8. *L187-189 Please add references, numbers or data to support your affirmation that the Farnebäck algorithm is robust for the complex morphology of cloud fields.**

We have included information on our selection of parameters for the Farnebäck algorithm which we found to provide the best balance between reliability and accuracy. We have also included data of our error estimates of the motion vectors calculated using the Farnebäck algorithm to demonstrate that the accuracy is sufficient for the use case presented in this paper. We find that the majority of optical flow vectors have a relative uncertainty of less than 0.1 for cold cloud scenes (BT <270K)

Changes regarding the use of the Farnebäck algorithm:

Line 233: "Although other optical flow algorithms may provide better accuracy in different circumstances (Baker et al., 2011), we have found that the ability of the Farnebäck algorithm to accept a range of parameters is important for detecting the motion of clouds across a wide range of scales. The choice of parameters used is a compromise between the ability to robustly detect flow vectors in areas of the image with low contrast between features (e.g. in the centre of anvil clouds), versus the fidelity of the motion vectors detected for small, high contrast features such as developing cores. The parameters we have chosen for the Farnebäck algorithm are shown in table 2. It should be noted that the values used for these parameters in the case of detecting the motion of DCCs is scale dependent. In particular, we find that the choice of window size is proportion to the time between subsequent images, and inversely proportional the spatial

resolution. The parameters given are for ABI imagery in the CONUS scan region, which has a temporal resolution of 5 minutes and the spatial resolution of 2 km."

Additional paragraph on the estimation of the uncertainty in the optical flow errors:

Line 246: "We estimate the uncertainty of the optical flow vectors by comparing the residual error in the location of features in subsequent images after the detected motion from optical flow is accounted for (see fig. 4d). We restrict the estimation of uncertainty to regions of clouds with brightness temperatures less than 270 K as these are the situations in which we see enough contrast with the background to detect optical flow motion vectors. We find that in the majority of cases that the relative uncertainty in the magnitude of the location offset versus the magnitude of the estimated optical flow vector is less than 10 %, with mean and median relative uncertainties of 15.0 % and 8.4 % respectively."

**9. In Fig. 9 the histograms are barely visible, please regenerate them with better choice of the axis limit**

After consideration, we have replaced figure 9 with a table containing the validation results as this presents the results in a clearer manner.

Small typos and notes:

L16 this framework to be applicable

L37 how satellites operating in the visible and IR

L40 large area

L43 the Geostationary Lightning Mapper

L45 Column mean radar reflectivity [...] shows

L52-53 The newest generation [...] offers

L156 GLM has the same field [...] L183 is used here for to L183 [...] tracking of DDCs . L197 based of of L200 its L204 include the effects L251 and edge-based L269 the detected anvil cloud begins L306 FAR of 0.27

Thank you for catching these errors, these have been corrected in the text.

**Reviewer 2**

**General comments**

The paper addresses detection and tracking of deep convective clouds using geostationary satellites. Various satellite instruments, including lightning flash detectors and radars are used for DCC detection, as well as various wavelengths ranging from visible to thermal infrared. The main novelty in the paper is the use of 'semi-Lagrangian' method to better account for cloud motion in a sequence of satellite images. Existing image processing methods can then be better applied for DCC cloud detection and tracking, leading to a significant improvement in correct classification of convective and non-convective clouds throughout their life-cycle.

The paper is well written, the methods seem sound, and in my opinion the results are interesting and valuable to the scientific community. The paper is clearly structured and cites the related previous research properly. The language is very clear, but there are a few typos here and there. However, some clarifications are needed. For example, I find that the typical challenges for the 'optical flow' and AMV techniques, and how they are tackled by the presented approach, are not sufficiently discussed in the methods part. Also, the limitations of the presented methods and their applicability to different situations should be briefly addressed. Some more detailed questions related to this are given in the 'specific comments' below.

I can recommend publication of the manuscript after a minor revision.

**Specific comments**

Page 3, line 38: The text says "Composite RGB images from a combination visible and near-IR channels" but the figure label says "visible RGB composite". The image does look like a regular (semi-)true-color satellite image. Please clarify.

The image is a semi-true colour composite using a combination of visible and near-IR channels (due to the lack of a green visible channel on ABI). We have changed the figure label to "visible/NIR composite" to make it consistent with the text.

Fig 2: What exactly is the cross section pictured here? Is the distance measured along a straight line from the mentioned origin to some direction (if so, please give also the end point of this line). Please clarify. It would help to show the position of the cross section on the map in Fig 1.

We have included a map showing the location of the cross section to figure 2.

Page 4, line 93: What does "over anvil area" mean here? Do you mean "overall/whole anvil area"? Please clarify.

Yes, this was meant to say "overall anvil area". This has been corrected in the text.

**Page 4, line 96: What is meant by 'active detection method' in this context?**

"Active detection method" refers to the use the growth/cooling of the DCC or other changes over time to detect DCCs. After consideration, we have decided that this term is unclear, so have replaced it with "growth-based detection" throughout the text, and used "thresholdbased detection" (previously "passive detection") to refer to the use of thresholds in brightness temperature or other fields for detecting DCCs.

Page 6, line 105: Please give the temporal extend of the data (year 2019) already here.

We have clarified both the temporal and spatial extent of the data used for both the cases shown in the figures throughout the paper (2018/06/19 18:00:00-00:00:00UTC) and for the validation period (all of 2018). The following has been added to the data section:

Line 142: "The case study shown in the figures throughout this paper is for a subset of the CONUS scan region centred at 83.7° W, 29.2° N, over the time period of 18:00:00 to

midnight UTC on the 19th June 2018. Validation was performed on a subset of the CONUS scan region from 114 to 76° W and 24 to 45° N over the entirety of 2018."

Page 8, line 149: I would expect that SWD would also be high for other cloud types, e.g. low level (optically thin) water clouds, which might cause false detections (unless combined with other methods, as in this case). Please clarify in the text.

Yes, this is correct. We have elaborated on the response of SWD to low level clouds and water vapour, and how this field cannot be used alone to differentiate between low- and high-level clouds, as follows:

Line 170: "The SWD is, however, also sensitive to low level clouds and low level water vapour concentrations, and so cannot be used alone to detect DCCs. It remains important to consider the SWD field due to the difficulty in separating anvil clouds from cirrus when using LW IR BT alone (Hong et al., 2005). By subtracting the SWD from the WVD field, we can reduce the sensitivity of our detection scheme to cirrus clouds, reducing the rate of erroneous detections. Further, adding the SWD field to WVD field can enhance the appearance of cirrus, enabling the detection of thin ice cloud associated with cirrus outflow and dissipating anvils."

**Page 8, section 3: Some more general comments on the methods section:**

I would suggest that the authors consider adding a very brief introduction to the methods section summarizing in a few lines the steps that are needed (e.g. with bullet points), to help the reader. The goal is to detect and track DCCs; existing image processing methods are used, but need to be refined in order to track DCCs, etc... 1) detection of AMV, 2) defining a semi-Lagrangian frame using the AMV, 3) careful selection of the channels used for anvil detection 4) applying the existing methods, but with the refined frame.

We thank the reviewer for this suggestion, and have added a brief outline of the method to the text as follows:

Line 188: "We present here a novel method for detecting and tracking both the growing cores and anvils clouds of DCCs, consisting of the following steps:

1. Ingest of LW IR BT fields from geostationary satellite imagery, including calculation of WVD and SWD fields from IR water vapour and LW IR window channels.

2. Calculation of optical flow vectors to be used as an a priori estimate of cloud motion for use in the semi-lagrangian framework.

3. Detection of growing DCC cores using cloud top cooling rate.

4. Detection of thick and thin anvil clouds associated with detected cores using a semilagrangian "3d" watershedding method. 5. Grouping of cores into multi-core systems, calculation of statistics and validation using lightning observations."

Typical problems in detection of specific atmospheric features such as plumes or clouds are related to the effect of the background, and specifically to the contrast between the background and the features being tracked. It is easier to detect and track high altitude cold clouds against a background of warm cloud free ocean than, say, against a cold land surface with possible low altitude cloud fields etc. Only one example is discussed in detail in the paper, but the method is applied over one year and over areas of varying conditions. Some of these issues are discussed in the text, but can the authors comment on the general applicability of the methods?

We find in general that the background does not affect the detection and tracking of highlevel clouds when using the WVD field, but it can affect the detection of low- and mid-level growing convection. We have added the following to the discussion of ABI channel selection regarding this:

Line 159: "Because both the WV channels are strongly absorbed by water vapour in the lower troposphere, the WVD field provides clear distinction between thick, high cloud and the background, whether that be land or ocean surface or low-level clouds. Müller et al. (2019) found that a threshold of -5 °C gave a high detection rate of anvil clouds. Furthermore, as the WVD values are relative to the lower stratosphere temperatures, this field is much less affected by location and meteorology than the LW IR channels. However, the WVD is still prone to the false detections of non-convective clouds when using a thresholding method as it cannot directly distinguish between thick, high-altitude cloud that are associated with deep convection and those that are not."

Likewise, methods based on tracking difference between images using cross correlation usually perform well when the features being tracked have sufficient contrast to the background, and also sufficiently variable texture. In other words, it would be easier to track the edges of an object with clear boundaries than, say, the central parts of a large cloud which have the same brightness temperature as the surrounding cloud. However, in Fig. 4 b) the optical flow vectors on different parts of the cloud agree surprisingly well. I would imagine this might be due to the mentioned use of 'increasingly smaller subsets of the image' in the Farnebäck method. If so, maybe this could be briefly elaborated.

The selection of parameters for the Farnebäck algorithm, particular the size and number of the pyramidal subsets, is important for being able to detect the motion of both large areas of clouds while also accurately tracking areas of high detail such as cloud edges and developing cores. The choice of window size, in particular, provides a compromise between smoothly estimating flow vectors is areas with reduced contrast (e.g. in the centre of anvil clouds) with larger windows vs providing higher fidelity to the estimated flow vectors around smaller,

higher contrast regions such as developing cores. We have included a table of the parameters we have used for the Farnebäck algorithm, which were selected to give a balance in capabilities between accurately detecting the motion of small objects and continually detecting motion within large anvil clouds with little contrast, and explained briefly why these values were chosen:

Line 228: "We use the Farnebäck algorithm (Farnebäck, 2003) to estimate optical flow vectors of observed DCCs. The Farnebäck algorithm calculates a 'dense' field of optical flow vectors, in which a flow vector is calculated for every pixel in the origin image which maps to its predicted location in the destination image. This calculation is performed by finding the minimum cross-correlation over increasingly smaller subsets of the image. This iterative approach allows flow vectors to be calculated to sub-pixel accuracy. In this framework, we have used the implementation of the Farnebäck algorithm from the OpenCV image processing package (Bradski, 2000). Although other optical flow algorithms may provide better accuracy in different circumstances (Baker et al., 2011), we have found that the ability of the Farnebäck algorithm to accept a range of parameters is important for detecting the motion of clouds across a wide range of scales. The choice of parameters used is a compromise between the ability to robustly detect flow vectors in areas of the image with low contrast between features (e.g. in the centre of anvil clouds), versus the fidelity of the motion vectors detected for small, high contrast features such as developing cores. Table 2 shows the parameters chosen for the Farnebäck algorithm for ABI imagery in the CONUS scan region, which has a temporal resolution of 5 minutes and the spatial resolution of 2 km. The values used for the window size parameter is scale dependent, proportional to the time between subsequent images and inversely proportional the spatial resolution."

**Page 8: section 3.1:**

Cross-correlation based methods typically have issues when applied to large features with homogeneous surface reflectance/brightness temperature. Are there any conditions where setting an AMV fails, or thresholds applied e.g. to minimum acceptable correlation? In other words, are there gaps in the AMV field that might affect your approach (or is AMV simply set to zero in uncertain cases)? Also, to clarify, are the AMV are constructed for the full image, not only clouds?

The AMV are constructed for the full image, not just clouds. The motion vectors default to zero in cases where the field in unclear. In general, this occurs for clear sky conditions, or for low clouds when there is not sufficient contrast between the clouds in the backgrounds. It should also be noted that the areas in which having the motion vectors is most important for accurately detecting and tracking the DCCs is at the edges and cores where there is the most contrast for calculating the optical flow. In cases where the cloud field is too homogeneous for optical flow vectors to be calculated, such as in large MCS cloud shields, the method behaves the same as the non-Lagrangian "3D" detection and tracking method of Fiolleau and Roca, which was designed for cases such as these, and so the method should still perform well here.

What are the major errors sources in the AMV calculations and can you estimate the resulting uncertainty? I would imagine that change in the cloud shape between the sequential images has an effect, and more notably the vertical motion can change the observed brightness temperatures, making it difficult to track the horizontal motion. Can you comment on the uncertainties?

We have included in fig. 4 the estimated uncertainty in the optical flow vectors based off of the residual error in the locations of objects in the field when advected between time steps. We find that the uncertainty is acceptably low (<10% in the majority of cases) for cold cloud scenes (brightness temperature <270K), particularly as most of the applications of the optical flow vectors within the presented methods are rounded to the nearest pixel. It should also be noted however that unlike for calculating AMVs for wind vectors, in this case it is actually desirable for the change in cloud shape to be included in the optical flow vectors as this helps both with the isolation of vertical growth, and with tracking individual DCCs between time steps. Furthermore, when calculating the optical flow vectors we normalise both fields in order to reduce the impact of changes in cloud top temperature on the ability to track cloud motion.

New discussion of uncertainty in the estimated optial flow vectors:

Line 246: "We estimate the uncertainty of the optical flow vectors by comparing the residual error in the location of features in subsequent images after the detected motion from optical flow is accounted for (see fig. 4d). We restrict the estimation of uncertainty to regions of clouds with brightness temperatures less than 270 K as these are the situations in which we see enough contrast with the background to detect optical flow motion vectors. We find that in the majority of cases that the relative uncertainty in the magnitude of the location offset versus the magnitude of the estimated optical flow vector is less than 10 %, with mean and median relative uncertainties of 15.0 % and 8.4 % respectively.

Two potential sources of uncertainty are the assumptions made by the Farnebäck algorithm that the feature being tracked remains the same size and intensity in subsequent images. For optical flow tracking using brightness temperature images of growing DCCs, neither of these assumptions are true. However, we have taken steps to reduce the impact of both these sources of uncertainty on the tracking algorithm. For the first of these assumptions, we find that in the case of small, fast moving DCCs – where the accuracy of the optical flow vectors is most important – the changes in size of the DCC is small compared to the overall motion, and so the uncertainty introduced is small. Comparably, for large DCCs where the changes in size may be large compared to the motion, the uncertainty introduces has less impact on the tracking algorithm. In the worst case scenario the estimated optical flow field will be zero, in which case the "3D" detection and tracking algorithm works in the same manner as that of Fiolleau and Roca (2013), which is suitable for use on these larger DCCs. To reduce uncertainty caused by the second assumption we normalise the range of brightness temperatures between subsequent images when estimating optical flow to reduce the change in brightness temperature observed for growing DCCs."

Page 9, Fig 4: Could you use a different color for the '+30min' dashed blue line for better separation form the solid blue line?

Which images are used to produce the shown atmospheric motion vectors (two successive images, I suppose, but at which time)?

Also, while the direction of the AMV arrows seems to agree with the change seen in contours, it is difficult to estimate if the magnitudes agree. From the caption I guess the 'reference arrow' at the bottom of the image corresponds to 1 pixel per frame i.e. 60 km/h in terms of velocity, but the length on the map is arbitrary. Also, the map does not have a length scale so it is difficult to say how much the contours change in the 30 min interval. Perhaps it would be more illustrative (in this particular case) if the length of the arrows was scaled so that it would correspond to the motion of a 'cloud particle' in the 30 minute interval, i.e. the particle would travel from the tail of the arrow to it's tip in this time period. Or, if this is technically too complicated, maybe briefly describe in the text how well these two methods agree on the magnitude of the motion.

*p.* 10, line 192: "By comparing the predicting flow vectors to the future evolution of the cloud field (dashed line), we can see that the algorithm correctly estimates the future evolution of the anvil cloud."

It is not entirely obvious what is meant here. From the text it sounds like you use three time steps: steps 1 and 2 to calculate 'predicting flow vectors', and steps 2 and 3 to illustrate the 'true evolution of the cloud field', apparently using just BT thresholds. Please clarify.

The previous comments are all regarding figure 4, so we have grouped our response together for these. It was difficult to produce this figure in a clear manner, and the result in the manuscript was a compromise. We thank the reviewer for their detailed comments, as this has helped us to reconsider the figure and make improvements to it. The optical flow vectors shown are calculated from subsequent images at 5 minute intervals. When plotting the brightness temperature with 5 minute intervals, the change in the cloud field was too small to be clearly seen. To address this, we instead plotted the brightness temperature field at 30 minute intervals to more clearly show the change over time. However, plotting the arrows scaled to 30 minute intervals results in a large overlap between arrows, making it difficult to see the motion. The arrow in the legend is scaled accurately to those in the plots, but is not very clear regardless. We have now revised this and made a number of changes to the figure and it's description in the text, which are described below:

We have changed the colour of the contours to improve clarity. The figure has been updated to include both the before and after images of the brightness temperature field. We have, in addition, plotted the magnitude of the optical flow vectors as a field behind the arrows so that the velocity of the flow vectors can be clearly seen. We have added a length scale to the axes of the plot. We have provided the colour scale of the flow vector velocity in km/hour, instead of pixels/frame, to more clearly compare against the plotted contours. Although it is important to note that the optical flow vectors are not a physical measurement of the velocity of the objects, it is clearer when comparing to the movement of the brightness temperature

field over time. We have also included the distribution of estimated uncertainty in the optical flow vectors to help show how accurately the optical flow method detects the evolution of the brightness temperature field.

We have also updated the text to better describe the figure and the optical flow vectors. In particular, we have removed the word "future" as this misleading as the optical flow vectors are not used to predict the subsequent motion of the cloud field shown, but instead derived between the initial and final fields.

*p.* 11, line 212: "Several implementations of common image processing operations have been developed using this Lagrangian convolution framework, including:"

Just to be sure, clarify what you have done in this work, and what has been developed and published before. Do you mean "We have applied this Lagrangian convolution framework to develop several new implementations of (existing) common image processing operations, including: ... "?

Yes, these are new implementations of existing image processing operations which have been adapted to use the Lagrangian framework. We have updated the text to the following:

Line 279: "We have developed new implementations of several existing image processing operations within the Lagrangian convolution framework, including: ..."

p. 13, line 253: Here you must mean WVD-SWD, instead of 'WVD field'? Please explain carefully at this point in the text why this subtraction is done. (It is to separate the edge of course, and described later in the text, but the way it is now written is confusing, and it takes several readings to understand what was actually done.)

Yes, this is correct. We have updated the text to make this clear and explain why the combination of fields is used.

**p. 13, line 262: Please give a brief motivation why this classification between thin and thick anvil clouds is made. I suppose it is related to the evaluation process.**

The classification into thick and thin anvils was made for two reasons. Firstly, when using the WVD alone for anvil detection, the edge of the anvil is detected when the anvil reaches an intermediate thickness (i.e. partially through the dissipating process). However, this state is difficult to distinguish from a mid-level cloud at the edge of the anvil. By using a combination of both the WVD and SWD fields, we can isolate only the thick, high-level anvil, and also the thin, dissipating region. This results in a better segmentation of anvil area, which, as you mention, is also relevant for the evaluation of the anvil detection. Secondly, we are also interested in researching the cirrus outflow from the detected anvils, and the

classification into thick and thin anvil allows us to identify what we believe to be thin cirrus regions.

p. 13, line 263: "By subtracting the SWD field from the WVD field, we are able to isolate only the thick regions of the anvil. By adding the SWD field to the WVD field, we are able to detect both the thick and thin anvil regions." Add a reference to Fig 7 a) here. (Would it be useful to also show the case SWD+WVD?) Please describe in more detail how the thin and thick regions are detected (by some thresholds?) in Fig 8. Please also consider if these combinations of VWD and SWD could be shown and briefly introduced already in connection with SWD and WVD definitions in section 2.1?

The SWD+WVD and detected edges of this field have been added to figure 7. We have added more detail to the text on how the thick and thin regions are detected using the edge-watershedding technique, which avoids the use of a fixed threshold by instead classifying the anvil extent based on detecting the edge of the cloud field. We have included a description of how the SWD field can be added/subtracted from the WVD field to enhance/reduce the appearance of thin, high-level clouds respectively to section 2.1:

Line 166: "The Split Window Difference (SWD), consisting of the clean IR window channel minus the

dirty IR window channel (fig. 3c), aids in the detection and separation of optically thin anvil cloud (including cirrus outflow) from optically thick anvil due to the difference in ice particle emissivity between these two channels (Heidinger and Pavolonis, 2009). As a result, this combination displays warm temperatures of around 10 K for thin, ice clouds, near 0 K for thick clouds, and approximately 5 K for clear skies due to the contribution of boundary layer water vapour. The SWD is, however, also sensitive to low level clouds and low level water vapour concentrations, and so cannot be used alone to detect DCCs. It remains important to consider the SWD field due to the difficulty in separating anvil clouds from cirrus when using LW IR BT alone (Hong et al., 2005). By subtracting the SWD from the WVD field, we can reduce the sensitivity of our detection scheme to cirrus clouds, reducing the rate of erroneous detections. Further, adding the SWD field to WVD field can enhance the appearance of cirrus, enabling the detection of thin ice cloud associated with cirrus outflow and dissipating anvils"

Update to section 3.4, discussing the selection of thresholds for the thick and thin anvil:

Line 317: "The region of anvil cloud associated with the growing convective clouds detected in the previous section is detected and tracked using an edge-base watershed segmentation approach. The edge-based approach to cloud detection avoids the use of a fixed threshold for anvil temperature, and so can detect a more accurate extent of the anvil cloud (Dim and Takamura, 2013). We define an upper threshold for the WVD field of -5 K, as used by Müller et al. (2018), and a lower threshold of -15 K, which we define as definite non-anvil cloud. Because the presence of thin cirrus outflow from the anvil clouds can make it difficult to determine the extent of the anvil cloud, we use the SWD field as described in section 2.1 to either remove or include the region of cirrus outflow in the detected anvil region. To detect the thick anvil cloud, we subtract the SWD field. In this case, the upper and lower threshold remain the same as the SWD field is approximately 0 K for thick, high clouds, and so has no effect on the temperature of these features. For detecting the thin anvil region, we add the SWD field and increase the value of both thresholds by 5 K to 0 K and -10 K respectively. This change is made to account for the effect of low level water vapour on the SWD field which gives a background value of approximately 5 K. Between these two thresholds, we have a region in which we are uncertain of the extent of the anvil cloud (see fig. 7). By applying a Sobel filter to detect the local gradient magnitude of the combined WVD/SWD field (Sobel, 2014), we detect the outer extent of the anvil cloud within this region where we see the greatest magnitude in the detected edges."

*p.* 14, Fig 7: Figures 3 and 7 show the same case, but slightly different areas are shown. Would it not be better to use the same format/area in all images?

We had previously shown different areas to highlight the features of interest in each plot, however on review we agree with your suggestion that it would be better to have the same area shown for all figures, and they have been updated.

p 14. Section 4: Evaluation is naturally focusing on the DCC cores, which are associated with the lightnings. This type of evaluation does not really work for the full extend of the anvil clouds, especially the thinner parts and dissipating phase. I suppose this is also the motivation for separating the detected clouds to thick and thin parts (see previous comments). I see that with the lightning data it is not possible to evaluate the extend of the thinner dissipating parts of the anvil clouds, which are naturally not associated with the lightnings. Including the thin parts to comparison in Fig. 9 would certainly degrade the statistics, and other forms of validation are required for the thin parts. As you mention, these dissipating parts may still have important role in e.g. climate studies. These points should be discussed in the text.

We have previously considered evaluating the detection of the thin anvil using lidar/radar instruments such as caliop and cloudsat, but decided that this went beyond the scope of this paper. We have now clarified in the text that lightning can only be used to validate the active cores of DCCs, and so is not suitable for validating the thin anvil extent:

Line 349: "It should be noted that lightning observations are only suitable for validating the detection of the thick anvil region, as lightning does not occur in the cirrus outflow. As a result, validation of the detection of the thin anvil region would require the use of other data

such as cloud profiling radar or lidar observations, and is not considered further in this paper."

Fig. 9: The labels for panels c) and f) are rather misleading, in that 'core and anvil' refers to a detected thick anvil region which can be associated with a DCC core at some point of it's lifetime. You should somehow emphasize also in the figure caption that b) corresponds to the 'old' simple threshold method while c) corresponds to the 'new' improved method which includes tracking of the DCCs.

We agree that the labels are not clear, and have changed "core only", "anvil only" and "core and anvil" to "growth based detection", "WVD threshold detection" and "Semi-lagrangian detection". We have also decided to replace figure 9 with a table showing the validation results, so these new labels correspond to the the table headings instead.

p. 16, line 290: How is the distance calculated? Is it zero if flash is within the detected anvil polygon, and the shortest distance to cloud edge outside the cloud polygon?

Yes, this is correct. We have added this to the text:

Line 367: "The distance between a GLM lightning flash and detected cloud region is defined as the distance between the flash and the nearest ABI pixel within that region, with GLM flashes that fall within a detected DCC given a distance of 0"

p. 17, line 296: What is a 'CONUS scan region'?

The CONUS scan region refers to the continental united states region of the GOES-16 scan pattern. We have added a description of the scan regions and the acronyms to the data section:

Line 126: "ABI operates in a flexible scan mode, imaging the continental US (CONUS) once every 5 minutes, the full disc every 10 minutes (15 minutes prior to April 2019), and two mesoscale regions of approximately 2500 by 2500 km every minute."

p. 17, Line 308: "We also evaluate the accuracy of detecting a fixed threshold of the WVD only to compare the detection of anvils without detecting growing cores, as used by Müller et al. (2018)."

Do you mean: "For comparison, we also evaluate the accuracy of detecting anvils only by a fixed threshold of the WVD without detecting growing cores, as in Müller et al. (2018)."?

Yes, this is correct. We have updated this in the text

p. 17, line 311: Maybe also highlight here the number of clouds detected using the simple method (n=323,618), and with the improved method developed in this work (n=14,717), showing how many non-convective clouds are included in the simple thresholds method.

This is an important point to raise, and we have included it in the text. We thank you for the suggestion.

Line 389: "Furthermore, the WVD threshold detection detects a much larger number of clouds (n=312,591) compared to either of the other detection methods, further indicating that a large number of non-convective clouds are detected using the threshold method on its own."

Note that the numbers have changes slightly as we found a few cases in which bad data was not being flagged as such in the ABI data, and so removed these days from the validation dataset. This has not had any impact on the calculated FAR/POD values